

# For catching-by-polymerization oligo purification: scalable synthesis of the precursors to the polymerizable tagging phosphoramidites

Yipeng Yin, Komal Chillar, Alexander Apostle, Bhaskar Halami, Adikari M. Dhananjani N. Eriyagama, Marina Tanasova and Shiyue Fang

Department of Chemistry, and Health Research Institute, Michigan Technological University, Houghton, MI, United States of America

## ABSTRACT

The catching-by-polymerization (CBP) oligodeoxynucleotide (oligo or ODN) purification method has been demonstrated suitable for large-scale, parallel, and long oligo purification. The authenticity of the oligos has been verified *via* DNA sequencing, and gene construction and expression. A remaining obstacle to the practical utility of the CBP method is affordable polymerizable tagging phosphoramidites (PTPs) that are needed for the method. In this article, we report scalable synthesis of the four nucleoside (dA, dC, dG and T) precursors to the PTPs using a route having five steps from inexpensive starting materials. The overall yields ranged from 21% to 35%. The scales of the synthesis presented here are up to 2.1 grams of the precursors. Because the syntheses are chromatography-free, further scaling up the syntheses of the precursors have become more feasible. With the precursors, the PTPs can be synthesized in one step using standard methods involving a chromatography purification.

## INTRODUCTION

Chemically synthesized oligodeoxynucleotides (oligos or ODNs) are typically purified with high-performance liquid chromatography (HPLC) and gel electrophoresis (*Yin et al., 2024b*). However, these methods have drawbacks such as the need of intense labor, and expensive or impossible to scale up. In addition, HPLC cannot purify oligos longer than 60-mer, and gel electrophoresis also has limitations in cases where the target long oligos are in a complex mixture. Many efforts have been made to address one or more of these problems, and various alternative methods have been reported. Examples include but are not limited to cartridge purification (*Semenyuk et al., 2006*; *Horn & Urdea, 1999*; *Ren, Osawa & Obika, 2024*), fluorous affinity purification (*Beller & Bannwarth, 2005*; *Pearson et al., 2005*), solid phase catch-and-release purification (*He et al., 2021*; *Grajkowski, Cieslak & Beaucage, 2016*; *Cawrse et al., 2023*), and biotin-streptavidin mediated affinity purification (*Fang & Bergstrom, 2003b*; *Franzini et al., 2015*; *Fang & Bergstrom, 2003a*). Among these efforts, our research group reported a method called catching-by-polymerization (CBP, Fig. 1) (*Yuan et*

Corresponding author
Shiyue Fang, shifang@mtu.edu

**Figure 1** **The catching-by-polymerization (CBP) oligo purification method.** Steps: (1) Synthesize ODN using a polymerizable tagging phosphoramidite (PTP, **1a–d**) to tag full-length ODN. Failure sequences are not tagged because they are capped with Ac$_2$O. (2) Cleave and deprotect ODN. (3) Co-polymerize tagged full-length ODN into a polyacrylamide gel. (4) Wash away failure sequences. (5) Cleave full-length ODN from polyacrylamide gel.

*al., 2012*; *Fang & Fueangfung, 2010*), for which two patents were awarded (*Fang, 2016*; *Fang, 2010*). The CBP method involves tagging the full-length sequences with a methacrylamide polymerizable tagging phosphoramidite (PTP, **1a-d**) in the last cycle of automated oligo synthesis. The failure sequences are capped with acetic anhydride, and thus are not tagged. After deprotection and cleavage under standard conditions, the crude product contains the desired full-length oligos having a methacrylamide polymerizable group (**2**), and failure sequences (**3**) and other impurities that do not have a polymerizable group. For purification, the crude product is subjected to co-polymerization under similar conditions widely used for preparing polyacrylamide gel for electrophoresis, which involves *N,N*-dimethylacrylamide, *N, N′*-methylenebis(acrylamide), and radical initiators. The full-length sequences are co-polymerized into the gel (**4**), and the impurities are removed by washing with water. Pure full-length oligos (**5**) are cleaved from the gel by breaking the trityl ether bond in **4** with an acid and extracted with water.

The CBP method has been demonstrated for large-scale (up to 60 μmol), parallel (up to 12 samples), and long oligo (up to 400-mer) purification (*Pokharel & Fang, 2016*; *Eriyagama et al., 2018*; *Yin et al., 2023*). It is remarkable that the method not only removes un-tagged impurities, but also concentrates the desired full-length sequence allowing isolation of minute quantities of full-length sequence from the complex mixture resulted from thousands of reactions required for the synthesis of 400-mer. The identity and purity of the oligos were analyzed with techniques such as HPLC, capillary electrophoresis, gel electrophoresis, MALDI MS, LC-MS, and enzymatic digestion assay, as well as cloning followed by Sanger sequencing. Construction of a gene using the long oligos and expression of the gene in *E. coli* have been successfully demonstrated. At this stage, one remaining obstacle for the practical utility of the CBP method, which requires about 25 μmol PTP (~30 mg) for each 1 μmol oligo synthesis, is the accessibility of the PTPs with affordable costs. This article is aimed at solving the problem. Portions of this article were previously published as a preprint (*Yin et al., 2024b*).

**Scheme 1  Chromatography-free synthesis of the precursors 10a-d to the polymerizable tagging phosphoramidites (PTPs) 1a–d.** Conditions: (A) *p*-MeOPhMgBr (2.5 eq), THF, 0 °C to rt, 3 h, 84%; (B) $Br(CH_2)_5CO_2CH_3$ (1 eq), $K_3PO_4$ (2 eq), acetone, DMSO, reflux, 12 h, 77%; (c) $[CH_2O(CH_2)_2NH_2]_2$ (3 eq), $H_2O$, 90 °C, 12 h, 79%; (D) methacrolyl chloride (1.1 eq), DIEA (5 eq), DCM, 0 °C to rt, 12 h, 90%; (E) TFAA (3 eq), DCM, 0 °C to rt, 2 h, remove volatiles, then deoxynucleosides T, dC$^{Ac}$, dA$^{Bz}$ or dG$^{iBu}$ (1.2 eq), DIEA (5 eq), DCM, pyridine, rt, 12 h, 77% (**1a**), 75% (**1b**), 51% (**1c**), 46% (**1d**); (F) (*i*Pr$_2$N)$_2$PO(CH$_2$)$_2$CN (1.5 eq), diisopropylammonium tetrazolide (1.5 eq), rt, 12 h, **1a** (82%). For yields of **1b–d**, see *Pokharel & Fang (2016)*.

## MATERIALS & METHODS

*Compound 6*: To a solution of 4-hydroxybenzophenone (10.0 g, 50.4 mmol, 1 eq) in dry THF (50 mL) at 0 °C under nitrogen was added 4-anisylmagnesium bromide (0.5 M in THF, 252.4 mL, 126 mmol, 2.5 eq) *via* a syringe. The mixture was stirred for 3 h while warming to rt. The excess 4-methoxyphenylmagnesium bromide was quenched with dry diisopropylamine (7.28 mL, 50.4 mmol, 1 eq). Dry Et$_2$O (500 mL) was added. The deprotonated anionic product was allowed to settle down at 0 °C under nitrogen overnight. The supernatant was removed *via* a cannula with its inflow end wrapped with a copper wire-secured cotton driven by a positive nitrogen pressure. To the precipitate was added water (10 mL) and EtOAc (200 mL). The content was poured into a separatory funnel containing saturated K$_2$CO$_3$ (100 mL), and partitioned. The aqueous layer was extracted with EtOAc (50 mL × 3). The combined organic layer was dried over anhydrous Na$_2$SO$_4$, filtered, concentrated to dryness, and dried under high vacuum. Compound **6**: 13 g, 84%, red foam, TLC (SiO$_2$) $R_f$ = 0.2 (3:1 hexanes/EtOAc). All compounds in this article are known (*Pokharel & Fang, 2016*). $^1$H NMR (500 MHz, CDCl$_3$) δ 3.82 (s, 3H), 6.77 (d, $J$ = 8.8 Hz, 1H), 6.85 (d, $J$ = 8.7 Hz, 2H), 7.12 (d, $J$ = 8.9 Hz, 2H), 7.19 (d, $J$ = 8.9 Hz, 2H), 7.28–7.32 (m, 7H); $^{13}$C NMR (126 MHz, CDCl$_3$) δ 55.3, 81.5, 113.2, 114.7, 114.8, 116.0, 127.1, 127.7, 127.8, 129.1, 129.3, 139.4, 147.2, 154.7, 158.5.

*Compound 7*: Compound **6** (10.0 g, 32.6 mmol, 1 eq), powdered potassium phosphate (13.9 g, 65.2 mmol, 2 eq), methyl-6-bromohexanoate (6.8 g, 32.6 mmol, 1 eq), DMSO (five mL), and dry acetone (50 mL) were combined. The mixture was stirred vigorously under nitrogen at reflux temperature for 12 h. Acetone was removed under reduced pressure, and DMSO was removed under vacuum from an oil pump. The residue was partitioned between

EtOAc (200 mL) and 5% $K_2CO_3$ (50 mL). The aqueous layer was extracted with EtOAc (20 mL × 3). The combined organic layer was dried over anhydrous $Na_2SO_4$, filtered, and concentrated to dryness. The product was dissolved in minimal DCM. Hexanes was added dropwise until the solution became cloudy. Minimal DCM was added to make a clear solution. The mixture was stored at −20 °C overnight. The supernatant was removed. The product in the form of a yellow oil was dried under high vacuum. Compound **7**: 11 g, 77%, light yellow oil, TLC ($SiO_2$) $R_f = 0.4$ (3:1 hexanes/EtOAc). $^1$H NMR (500 MHz, $CDCl_3$) δ 1.49–1.55 (m, 2H), 1.69–1.85 (m, 4H), 2.37 (t, $J = 7.5$ Hz, 2H), 3.69 (s, 3H), 3.82 (s, 3H), 3.97 (t, $J = 6.4$ Hz, 2H), 6.84 (t, $J = 9.0$ Hz, 1H), 7.17–7.20 (m, 4H), 7.28–7.34 (m, 9H); $^{13}$C NMR (126 MHz, $CDCl_3$) δ 24.6, 25.6, 28.9, 33.9, 51.5, 55.2, 67.5, 81.3, 113.1, 113.7, 127.0, 127.7, 129.1, 139.3, 139.4, 147.3, 158.1, 158.5, 174.0; HRMS (ESI) *m/z* calcd for [M + Na]$^+$ $C_{27}H_{30}NaO_5$ 457.1991, found 457.1984.

*Compound 8*: Compound **7** (10.0 g, 23.03 mmol, 1 eq), 2, 2′-(ethylenedioxy)bis(ethylamine) (10.2 g, 69.09 mmol, 10.1 ml, 3 eq), and water (0.5 mL) were combined. The mixture was stirred vigorously at 90 °C for 12 h. After cooling to rt, the content was transferred into a separatory funnel, and partitioned between 10% $K_2CO_3$ (50 mL) and DCM (50 mL). The aqueous layer was extracted with DCM (20 mL × 3). The combined organic layer was dried over anhydrous $Na_2SO_4$, filtered, concentrated to dryness, and dried under high vacuum. Compound **8**: 10 g, 79%, light yellow gel, TLC ($SiO_2$) $R_f = 0.3$ (5:2:2:1 $Et_2O$/MeCN/MeOH/$Et_3N$). $^1$H NMR (500 MHz, $CD_3OD$) δ 1.46–1.53 (m, 2H), 1.65–1.80 (m, 3H), 2.23 (t, $J = 7.4$ Hz, 2H), 2.79 (t, $J = 5.2$ Hz, 2H), 3.37 (t, $J = 5.5$ Hz, 2H), 3.50–3.55 (m, 6H), 3.60 (s, 3H), 3.75 (s, 3H), 3.94 (t, $J = 6.3$ Hz, 2H), 6.81 (t, $J = 7.7$ Hz, 2H), 7.11–7.14 (m, 4H), 7.21–7.26 (m, 9H); $^{13}$C NMR (126 MHz, $CD_3OD$) δ 25.2, 28.6, 35.5, 38.8, 40.5, 54.3, 67.3, 69.2, 69.9, 71.6, 80.9, 112.5, 113.5, 126.4, 127.2, 127.8, 129.0, 139.8, 139.9, 148.0, 158.0, 158.5, 174.7; HRMS (ESI) *m/z* calcd for [M + H]$^+$ $C_{32}H_{43}N_2O_6$ 551.3121, found 551.3126.

*Compound 9*: To a solution of compound **8** (10.0 g, 18.1 mmol, 1 eq) and DIEA (11.72 g, 90.6 mmol, 16.74 mL, 5 eq) in dry DCM (40 mL) was added the solution of methacrolyl chloride (2.08 g, 19.9 mmol, 1.94 mL, 1.1 eq) in dry DCM (10 mL) slowly *via* a cannula along the wall of the flask at 0 °C under nitrogen while stirring the mixture vigorously. After addition, stirring was continued for 12 h while warming to rt gradually. The content was poured into a separatory funnel containing 10% $K_2CO_3$ (50 mL), and partitioned. The aqueous layer was extracted with DCM (20 mL × 3). The combined organic layer was dried over anhydrous $Na_2SO_4$, filtered, and concentrated to dryness. The product was dissolved in minimal THF containing 1% DIEA. Hexanes was added dropwise until the solution became cloudy. A few drops of THF was added to make the solution clear. The mixture was stored at −10 °C overnight. The supernatant was removed. The product in the form of a light yellow oil was dried under high vacuum. Compound **9**: 10 g, 90%, white foam, TLC ($SiO_2$) $R_f = 0.4$ (3:2 acetone/hexanes 5% $Et_3N$). $^1$H NMR (500 MHz, $CDCl_3$) δ 1.43–1.50 (m, 2H), 1.64–1.71(m, 2H), 1.73–1.79 (m, 2H), 1.94 (s, 3H), 2.17 (t, $J = 7.5$ Hz, 2H), 3.39–3.57 (m, 12H), 3.77 (s, 3H), 3.92 (t, $J = 6.3$ Hz, 2H), 5.29 (d, $J = 14.3$ Hz, 2H), 5.69 (s, 1H), 6.27 (s, 1H), 6.45 (s, 1H), 6.80 (t, $J = 10.3$ Hz, 2H), 7.13–7.17 (m, 4H), 7.22–7.28 (m, 9H); $^{13}$C NMR (126 MHz, $CDCl_3$) δ 18.7, 25.4, 25.8, 29.0, 36.5, 39.1, 39.4,

55.3, 67.6, 69.8, 70.0, 70.1, 70.2, 81.4, 113.1, 113.6, 119.7, 127.0, 127.8, 129.2, 139.5, 139.9, 147.5, 158.0, 158.5, 168.6, 173.2; HRMS (ESI) $m/z$ calcd for $[M + Na]^+$ $C_{36}H_{46}N_2NaO_7$ 641.3203, found 641.3209.

*Compound 10a*: To a solution of **9** (2.0 g, 3.23 mmol, 1 eq) in dry DCM (20 mL) at 0 °C under nitrogen was added trifluoroacetic acid anhydride (TFAA, 2.03 g, 9.70 mmol, 1.34 mL, 3 eq) *via* a syringe dropwise. After stirring for 2 h while warming to rt, a deep red solution was formed. DCM was evaporated on a rotary evaporator under reduced pressure provided by a water aspirator *via* a drying tube containing Drierite. The trifluoroacetic acid side product and the excess TFAA were removed under high vacuum. After drying under high vacuum for 30 min, the intermediate was dissolved in dry pyridine (10 mL) and added *via* a cannula to a solution of thymidine (0.94 g, 3.88 mmol, 1.2 eq) and DIEA (2.09 g, 16.17 mmol, 2.82 mL, 5 eq) in dry DCM (10 mL) and pyridine (10 mL). After stirring overnight, the content was poured into a separatory funnel containing 10% $K_2CO_3$ (50 mL), and partitioned. The aqueous layer was extracted with DCM (20 mL × 3). The combined organic layer was dried over anhydrous $Na_2SO_4$, filtered, and concentrated to dryness. The residue was dissolved in minimal DCM containing 1% DIEA. Hexanes was added dropwise until the solution became cloudy. Minimal DCM was added to make the solution clear. The solution was stored at −10 °C overnight. The supernatant was removed. The light yellow solid product was dried under high vacuum. Compound **10a**: 2.1 g, 77%, white foam, TLC (SiO$_2$) $R_f = 0.2$ (3:2 acetone/hexanes 5% Et$_3$N). $^1$H NMR (500 MHz, CDCl$_3$) δ 1.41 (s, 3H), 1.47–1.77 (m, 6H), 1.95 (s, 3H), 2.20 (t, $J = 7.4$ Hz, 2H), 2.30–2.41 (m, 2H), 3.37–3.62 (m, 10H), 3.92 (t, $J = 6.2$ Hz, 2H), 4.08 (s, 1H), 4.57 (s, 1H), 5.32 (s, 2H), 5.71 (s, 2H), 6.42 (t, $J = 6.8$ Hz, 1H), 6.79–6.84 (m, 4H), 7.14–7.41 (m, 9H), 7.63 (s, 1H); $^{13}$C NMR (126 MHz, CDCl$_3$) δ 11.8, 18.6, 25.3, 25.7, 28.9, 36.4, 39.1, 39.4, 40.9, 63.6, 67.6, 69.6, 69.8, 70.1, 70.2, 72.2, 81.3, 84.7, 86.3, 86.8, 111.0, 113.1, 113.2, 113.6, 113.7, 119.7, 127.1, 127.9, 128.1, 129.1, 130.0, 130.1, 135.3, 135.9, 139.4, 139.8, 144.2, 147.4, 150.5, 158.1, 158.6, 164.1, 168.7, 173.2; MS (ESI) $m/z$ calcd $C_{46}H_{58}N_4O_{11}Na$ $[M + Na]^+$ 865.4000, found 865.3998.

*Compound 10b*: Synthesized using the same procedure for **10a**. Compound **9** (2.0 g, 3.23 mmol, 1 eq), DCM (20 mL), TFAA (2.03 g, 9.70 mmol, 1.34 mL, 3 eq), $N$ 4-acetyl-2′-deoxycytidine (1.04 g, 3.88 mmol, 1.2 eq), DIEA (2 g, 15.55 mmol, 2.8 mL, 5 eq), DCM (10 mL), and pyridine (10 mL) were used. The product was purified by partition and precipitation under the conditions described for **10a**. Compound **10b**: 2.1 g, 75%, white foam, TLC (SiO$_2$) $R_f = 0.2$ (3:2 acetone/hexanes with 5% Et$_3$N). $^1$H NMR (500 MHz, CDCl$_3$) δ 1.44–1.78 (m, 6H), 1.95 (s, 3H), 2.21 (s, 2H), 2.22 (s, 3H), 2.76–3.28 (m, 2H), 3.38–3.58 (m, 10H), 3.59 (s, 4H), 3.93 (t, $J = 7.5$ Hz, 2H), 4.17 (s, 1H), 4.49–4.52 (m, 1H), 5.31 (s, 2H), 5.69 (s, 2H), 6.22–6.44 (m, 2H), 6.80–6.85 (m, 4H), 7.18–7.31 (m, 8H), 7.40 (d, $J = 10$ Hz, 1H), 8.22–8.24 (m, 1H); $^{13}$C NMR (126 MHz, CDCl$_3$) δ 18.6, 24.8, 25.3, 25.7, 28.9, 36.5, 39.1, 39.3, 42.0, 55.2, 62.7, 67.5, 69.6, 69.8, 70.1, 70.2, 70.6, 86.5, 86.8, 87.2, 96.5, 113.2, 113.7, 119.7, 127.0, 127.9, 130.0, 135.2, 135.4, 139.7, 144.2, 144.5, 155.4, 158.1, 158.6, 162.6, 168.6, 173.2; HRMS (ESI) $m/z$ calcd for $C_{47}H_{59}N_5O_{11}Na$ $[M + Na]^+$ 892.4, found 892.5.

*Compound 10c*: Synthesized using the same procedure for **10a**. Compound **9** (1.0 g, 1.61 mmol, 1 eq), DCM (20 mL), TFAA (1.01 g, 4.85 mmol, 0.67 mL, 3 eq), *N* 6-benzoyl-2′-deoxyadenosine (0.687 g, 1.94 mmol, 1.2 eq), DIEA (1.04 g, 8.08 mmol, 1.41 mL, 5 eq), DCM (10 mL), and pyridine (10 mL) were used. The product was purified by partition and precipitation under the conditions described for **10a**. Compound **10c**: 0.79 g, 51%, white foam, TLC (SiO$_2$) $R_f$ = 0.1 (3:2 acetone/hexanes 5% Et$_3$N). $^1$H NMR (500 MHz, CDCl$_3$) $\delta$ 1.45–1.76 (m, 6H), 1.94 (d, $J$ = 10 Hz, 3H), 2.19 (t, $J$ = 7.5 Hz, 2H), 2.56–2.86 (m, 2H), 3.38–3.92 (m, 15H), 4.21 (s, 2H), 4.68 (s, 1H), 5.29 (s, 1H), 5.69 (s, 1H), 6.17–6.50 (m, 1H), 6.74–6.82 (m, 4H), 7.13–7.39 (m, 9H), 7.46–7.57 (m, 3H), 8.01 (d, $J$ = 10 Hz, 2H), 8.10 (s, 1H), 8.14 (s, 1H), 8.69 (s, 1H); $^{13}$C NMR (126 MHz, CDCl$_3$) $\delta$ 18.6, 22.9, 23.9, 25.3, 25.8, 28.9, 29.7, 36.4, 39.1, 39.4, 63.7, 67.5, 69.6, 70.1, 72.1, 81.3, 84.8, 86.5, 113.0, 113.1, 113.5, 113.6, 119.7, 126.9, 127.7, 127.9, 128.0, 128.7, 129.1, 129.5, 129.9, 130.0, 132.6, 135.4, 135.7, 139.5, 139.8, 141.5, 144.5, 147.5, 158.0, 158.4, 168.5, 173.1; HRMS (ESI) *m/z* calcd for C$_{53}$H$_{61}$N$_7$O$_{10}$Na [M + Na]$^+$ 978.4, found 978.5.

*Compound 10d*: Synthesized using the same procedure for **10a**. Compound **9** (1.0 g, 1.61 mmol, 1 eq), DCM (20 mL), TFAA (1.01 g, 4.85 mmol, 0.67 mL, 3 eq), *N* 2-isobutyryl-2′-deoxyguanosine (0.654 g, 1.94 mmol, 1.2 eq), DIEA (1.04 g, 8.08 mmol, 1.41 mL, 5 eq), DCM (10 mL), and pyridine (10 mL) were used. The product was purified by partition and precipitation under the conditions described for **10a**. Compound **10d**: 0.7 g, 46%, white foam, TLC (SiO$_2$) $R_f$ = 0.2 (3:2 acetone/hexanes 5% Et$_3$N). $^1$H NMR (500 MHz, CDCl$_3$) $\delta$ 1.18 (t, $J$ = 5 Hz, 6H), 1.47–1.75 (m, 6H), 1.95 (s, 3H), 2.25 (t, $J$ = 7.5 Hz, 2H), 2.47–2.72 (m, 3H), 3.27–3.58 (m, 10H), 3.59 (s, 4H), 3.86 (s, 2H), 4.19 (s, 1H), 4.74 (s, 1H), 5.32 (s, 1H), 5.72 (s, 1H), 6.25 (t, $J$ = 5 Hz, 1H), 6.57–6.82 (m, 4H), 7.13–7.38 (m, 9H), 7.79 (s, 1H); $^{13}$C NMR (126 MHz, CDCl$_3$) $\delta$ 18.6, 18.9, 22.8, 25.2, 28.8, 34.4, 35.9, 38.8, 39.2, 39.4, 64.2, 67.5, 69.6, 69.7, 70.1, 70.2, 81.3, 84.1, 86.3, 86.6, 113.0, 113.5, 119.9, 121.0, 126.8, 127.7, 127.8, 128.0, 129.1, 129.2, 129.4, 129.9, 130.1, 139.8, 148.4, 157.9, 158.4, 168.7, 173.6; HRMS (ESI) *m/z* calcd for C$_{50}$H$_{63}$N$_7$O$_{11}$Na [M + Na]$^+$ 960.5, found 960.7.

*Compound 1a*: To a solution of compound **10a** (500 mg, 0.59 mmol, 1 eq) in dry ACN was added diisopropylammonium tetrazolide (0.15 g, 0.88 mmol, 1.5 eq) and 2-cyanoethyl $N,N,N′,N′$-tetraisopropylphosphorodiamidite (0.26 g, 0.88 mmol, 1.5 eq) at rt under nitrogen. After stirring overnight, the mixture was concentrated to dryness. The product was purified by dissolving in the solvent mixture of acetone/hexanes 3:1 with 5% Et$_3$N, loading onto a column (SiO$_2$), and eluting with the same solvent mixture. The product was given as a white foam upon drying under high vacuum: 0.51 g, 82%; white foam; TLC $R_f$ = 0.2 (SiO$_2$, acetone/hexanes 3:2, 5% Et$_3$N); $^1$H NMR (500 MHz, CDCl$_3$) $\delta$ 1.03–1.05 (m, 3H), 1.15–1.17 (m, 3H), 1.39–1.79 (m, 6H), 1.95 (s, 3H), 2.16 (s, 3H), 2.20 (t, $J$ = 7.5 Hz, 2H), 2.30–2.63 (m, 5H), 3.30–3.62 (m, 10H), 3.93 (s, 2H), 4.18 (s, 1H), 4.67 (s, 1H), 5.31 (s, 1H), 5.70 (s, 1H), 6.32–6.52 (m, 1H), 6.78–6.84 (m, 4H), 7.11–7.40 (m, 9H), 7.64 (s, 1H); $^{13}$C NMR (126 MHz, CDCl$_3$) $\delta$ 11.1, 11.6, 18.6, 24.5, 25.2, 25.7, 28.9, 36.3, 39.0, 39.4, 45.9, 55.1, 58.1, 63.1, 67.5, 70.0, 73.6, 86.7, 111.2, 113.2, 113.7, 117.4, 119.5, 127.1, 127.9, 130.0, 135.3, 139.9, 144.1, 150.4, 158.0, 158.6, 164.0, 168.5, 173.0; $^{31}$P NMR (202 MHz, CDCl$_3$) $\delta$ 148.4, 148.7; MS (ESI) *m/z* calcd for [M + Na]$^+$ C$_{55}$H$_{75}$N$_6$NaO$_{12}$P 1065.5, found 1065.4.

## RESULTS AND DISCUSSION

Because phosphoramidite monomers are typically purified with chromatography to ensure high coupling yields for oligo synthesis, and alcohol phosphitylation reactions are well known, this work was focused on the chromatography-free synthesis of the precursors **10a–d** to the phosphoramidites (Scheme 1), and the conversion of **10a–d** to **1a–d**, which was reported previously (*Pokharel & Fang, 2016*), was not optimized to remove chromatography purification in the present work. The synthetic route includes six steps starting from the inexpensive 4-hydroxybenzophenone and 4-anisylmagnesium bromide. Because our goal was to remove all chromatography purifications so that the synthesis is scalable, efforts were made to ensure the reactions to be as clean as possible. After the reactions, various conditions including different partition and precipitation conditions were screened to purify the products. Despite the perceived difficulty posed by the elongated molecular chains, which hinders crystallization, our perseverance in exploring various synthesis and purification methods yielded the target compounds with good purity without chromatography purification.

The first step was to convert 4-hydroxybenzophenone to the trityl alcohol **6** (Scheme 1). Because 4-hydroxybenzophenone is a solid, and cannot be removed by evaporation during product purification, we used excess 4-anisylmagnesium bromide to drive the reaction to completion. However, we found that removal of the anisole side product by evaporation was not easy either even though its boiling point is only 154 °C. To solve this problem, the reaction was quenched with dry diisopropylamine instead of water. Under these conditions, **6** remained anionic, which was precipitated readily from the THF solution by diethyl ether. Anisole remained in the supernatant, which was removed by filtration under inert atmosphere. Optionally, to ensure complete removal of anisole, the product can be washed with dry diethyl ether under inert atmosphere until TLC indicates absence of anisole. However, in most cases, after partition between saturated potassium carbonate and ethyl acetate, pure product could be obtained without additional dry diethyl ether washes. The reaction was conducted at scales up to 13 grams of product. Yields around 84% were obtained consistently. Images of product, TLC, and proton and carbon NMR spectra for compounds (**6–10**) in this article are provided in Data S1.

For the synthesis of compound **7**, one challenge was that both starting materials could not be removed by evaporation if they were not consumed completely. For this reason, equal molar quantities of them were used. After screening several solvents and bases, we found that using potassium phosphate as the base, and acetone and suitable amounts of DMSO as the solvent could give a clean reaction. After partitioning between 5% potassium carbonate and ethyl acetate, the product, which had good purity as indicated by TLC, was obtained by precipitation from DCM by hexanes. The reaction was conducted at scales up to 11 grams of product. The yields were around 77%. The product was a light yellow oil and could not be foamed.

Synthesis of compound **8** requires mono-acylation of a symmetric diamine, which had been challenging especially if pure product needs to be obtained without chromatography because symmetric diacylated products were reported to be favored over the mono-acylated
ones in the literature (*Schwabacher et al., 1998*; *Jacobson, Makris & Sayre, 1987*; *Zhang et al., 2003*). Fortunately, a few years ago, it was reported that in the presence of a small amount of water, with various esters and acid chlorides as the acylation agents, mono-acylation could be achieved with high selectivity (*Pappas, Zhang & Fang, 2010*; *Pappas et al., 2009*; *Tang & Fang, 2008*). Based on those discoveries, we simply heated compound **7** and 2, 2′-(ethylenedioxy)bis(ethylamine) at about 90 °C in the presence of suitable amounts of water as determined previously (*Pappas, Zhang & Fang, 2010*; *Pappas et al., 2009*; *Tang & Fang, 2008*). The reaction could complete in 12 h, and no diacylated product was detectable by TLC. The excess diamine is water soluble and was removed by partition. The reaction was carried out at scales up to 10 grams of product, and pure **8** was obtained in 79% yield without any need for chromatography purification.

The acylation of compound **8** with methacryolyl chloride to yield **9** initially appeared straightforward; however, we encountered unforeseen challenges during the process. The reaction had not been clean enough for non-chromatographic purification under a variety of conditions. To avoid the potential side reaction between the acid chloride and the trityl alcohol moiety of **8**, which has the likelihood to generate the highly reactive trityl chloride, we tested to use less reactive acylation agents such as methyl methacrylate, phenyl methacrylate, pentafluorophenyl methacrylate, and methacrylic anhydride. However, the results were not much better than the use of methacryloyl chloride. In the end, we were able to make the reaction clean enough for non-chromatographic purification of product by adding a methacryloyl chloride solution in DCM (as opposed to the pure reagent) *via* a cannula slowly along the cooled wall of the reaction flask at 0 °C while stirring the reaction mixture vigorously. To ensure complete consumption of **8**, we still used a slight excess of methacryloyl chloride, which means that a small amount of trityl chloride was possibly formed. However, that was after all the amino groups were acylated, and the trityl chloride was probably hydrolyzed during subsequent aqueous workup. The product was purified by partition between 10% potassium carbonate and DCM, and precipitation from 1% DIEA in THF by hexanes. It is noted that including DIEA was important. Otherwise, the results could be inconsistent as TLC gave additional spots aside from the product. Evidently, the role of DIEA is to prevent ionization of the trityl alcohol.

The final step for the synthesis of the precursors **10a–d** was to attach **9** to deoxynucleoside derivatives (T, dC$^{Ac}$, dA$^{Bz}$ and dG$^{iBu}$) *via* the formation of a trityl ether bond at its 5′-OH. This reaction was typically carried out by converting the trityl alcohol to trityl chloride using acetyl chloride (*Pearson et al., 2005*). However, we found that the use of a method reported earlier by *Shahsavari et al. (2016)* had significant advantages (*Tang & Fang, 2008*). This method activates the trityl alcohol to the more reactive trityl cation using trifluoroacetic anhydride, and reacts the cation *in situ* with an alcohol to form the trityl ether bond. The method has been demonstrated to be efficient for the tritylation of not only the less hindered primary alcohol but also the more hindered secondary alcohols without using any additional activating agents such as silver salts. Using this method, the reactions for the synthesis of compounds **10a–d** were fairly clean as indicated by TLC (see Supporting Information for TLC images). Three notes may be beneficial for others to repeat the reaction. One is that during the formation of the trityl cation, it is critical to minimize the

exposure of the cationic intermediate to moisture. For this, when removing excess TFAA, solvent DCM and side product TFA on a rotary evaporator, a drying tube containing Drierite was installed between the water aspirator and rotary evaporator to minimize diffusion of moisture into the flask. In addition, removing residue volatiles under high vacuum was also conducted quickly to minimize exposure to moisture. The other note is that slightly excess nucleosides should be used to ensure near complete consumption of compound **9**. The reason is that the nucleosides are water soluble and can be removed by partition between an organic solvent and water, while compound **9** can be difficult to remove without using chromatography. The third note is that during precipitation, like the precipitation of compound **9**, it is important to include a small amount of DIEA in the solution. Otherwise, the results would vary from time to time due to the instability of the trityl ether bond under even slightly acidic conditions. The products (**10a–d**) could not crystallize. However, they became a foam upon applying high vacuum (see images of products in the Supporting Information), which was beneficial for thorough drying of them and for the next highly moisture sensitive phosphitylation reaction for converting them to the target polymerizable tagging phosphoramidites (PTPs) **1a–d**. For compounds **10a–d**, besides images of product, TLC, and proton and carbon NMR as provided for other compounds, those of MS are also provided in Data S1.

The average overall yield for the synthesis of **10a–d** from 4-hydroxybenzophenone is 29%. Earlier, we synthesized **10a–d** from **6** using a procedure that required four column chromatography purifications (*Pokharel & Fang, 2016*). The average overall yield was 41%. For comparison, the average overall yield for the synthesis of **10a–d** from **6** using the current non-chromatographic method is 34%, which is slightly lower than the earlier procedure. The reason for this is that during purification by precipitation, some products remained in the supernatant, and were lost. One may consider recovering them, but typically that is not economically worthwhile. The major advantage of the current method is that it does not require any chromatography. The advantage is significant because it makes the procedure scalable with acceptable costs. To illustrate this point, we can use the scaling up of a 10-gram product purification to a 1,000-gram product purification as an example. In the case of purification by chromatography, depending on the complexity of the crude product, purifying 10 grams of product requires about 200 to 500 grams of silica gel and about 2,000 to 5,000 grams of solvent, as well as one day of an experienced chemist's time. In addition, instrumentation and the energy for solvent evaporation also cost money. Based on these numbers, it is evident that the cost for scaling up the purification to 1,000 grams can be prohibitive under most circumstances. In contrast, in the case of purification by precipitation, scaling up a 10-gram product purification to a 1,000-gram product purification would induce very little additional costs.

To illustrate that the compounds **10a–d** synthesized without chromatography purification can be reliably used for the synthesis of the PTPs **1a–d**, the phosphitylation reaction for **10a** reported earlier was repeated without attempting to remove chromatography. The reason for not making such an attempt is that **1a–d** are the final compounds that are to be used for oligo synthesis, and purification with chromatography or HPLC of phosphoramidites is a common practice in the industry of oligo synthesis

reagent production. Using **10a** that was not purified with chromatography as the starting material, the reaction was clean (see TLC image in the Supporting Information). The $^1$H, $^{13}$C and $^{31}$P NMR spectra for **1a** are in the Supporting Information. As it is usually the case, the $^1$H and $^{13}$C spectra are very complex due to the additional peaks from different diastereoisomers and coupling with the phosphorus atom as well as the large size of the compounds. However, the $^{31}$P NMR spectrum is clean. The PTP **1a** synthesized using the present method had been successfully used for CBP purification of 400-mer oligos (*Yin et al., 2023*), and more recently the 800-mer GFP gene and the 1,728-mer Φ29 DNA polymerase gene (*Yin et al., 2024a*). Earlier, PTP **1b–d** were also synthesized, and were successfully used for purification of long oligos as well (*Pokharel & Fang, 2016*). These long oligos would otherwise be impossible to purify without the use of these PTPs and the CBP method.

## CONCLUSIONS

In summary, to make the catching-by-polymerization (CBP) oligo purification method practically useful, a scalable method for the synthesis of the precursors (**10a–d**) to the PTPs **1a–d** has been developed. The synthetic route involves six steps. Even though the compounds and the intermediates in the synthesis were not crystallizable, which makes non-chromatographic product purification less likely, by careful optimization of reaction conditions to make the reactions as clean as possible and by searching suitable precipitation conditions, we were able to synthesize all the compounds without any chromatography. Compound **10a** was converted to the target PTP **1a**, which had been successfully used for CBP purification of genes with up to 1,728 nucleotides consecutively synthesized on an automated synthesizer (*Yin et al., 2023*; *Yin et al., 2024a*). The use of the CBP method for synthetic RNA purification is being actively pursued. The work reported here paves the way for large-scale production of the PTPs at affordable costs, removing a critical barrier to the practical utility of the CBP oligo purification method.

### Funding
This work was supported by the National Science Foundation (1954041) and the National Institutes of Health (GM109288), as well as student fellowships including the Robert and Kathleen Lane Endowed Fellowship (Yipeng Yin, Komal Chillar, Alexander Apostle), the David and Valeria Pruett Fellowship (Adikari M.D.N. Eriyagama), the Fleming-Skochelak Fellowship (Komal Chillar), the MSGC fellowship (Alexander Apostle), the HRI Fellowship (Komal Chillar, Alexander Apostle), and the Doctoral Finishing Fellowship (Yipeng Yin, Komal Chillar). The funders had no role in study design, data collection and analysis, decision to publish, or preparation of the manuscript.

### Grant Disclosures
The following grant information was disclosed by the authors:

National Science Foundation: 1954041.
National Institutes of Health: GM109288.
Robert and Kathleen Lane Endowed Fellowship.
David and Valeria Pruett Fellowship.
Fleming-Skochelak Fellowship.
HRI Fellowship.
Doctoral Finishing Fellowship.

## Competing Interests

Michigan Tech holds two patents related to the catching-by-polymerization oligo purification method. Shiyue Fang. Purification of synthetic oligonucleotides. US9243023B2. January 26, 2016. Shiyue Fang. Purification of synthetic oligomers. US7850949B2. December 14, 2010.

## Author Contributions

- Yipeng Yin conceived and designed the experiments, performed the experiments, analyzed the data, prepared figures and/or tables, authored or reviewed drafts of the article, and approved the final draft.
- Komal Chillar performed the experiments, authored or reviewed drafts of the article, and approved the final draft.
- Alexander Apostle performed the experiments, authored or reviewed drafts of the article, and approved the final draft.
- Bhaskar Halami performed the experiments, authored or reviewed drafts of the article, and approved the final draft.
- Adikari M. Dhananjani N. Eriyagama performed the experiments, authored or reviewed drafts of the article, and approved the final draft.
- Marina Tanasova performed the experiments, authored or reviewed drafts of the article, and approved the final draft.
- Shiyue Fang conceived and designed the experiments, performed the experiments, analyzed the data, prepared figures and/or tables, authored or reviewed drafts of the article, and approved the final draft.

## Patent Disclosures

The following patent dependencies were disclosed by the authors:

Shiyue Fang. Purification of synthetic oligonucleotides. US9243023B2. January 26, 2016.

Shiyue Fang. Purification of synthetic oligomers. US7850949B2. December 14, 2010.

## Data Availability

The raw data are in the Supplementary File.

## Supplemental Information

Supplemental information for this article can be found online at http://dx.doi.org/10.7717/peerj-ochem.12#supplemental-information.

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
