# Peer review of "For catching-by-polymerization oligo purification: scalable synthesis of the precursors to the polymerizable tagging phosphoramidites"

_PeerJ Organic Chemistry, doi:10.7717/peerj-ochem.12_

## Round 0.1 · original submission · Major Revisions

Since the main focus of this manuscript is to offer an alternative route for the synthesis of modified phosphoramidite, I strongly agree with reviewer #2 that additional evidence for purity is required for the final compounds (analytical HPLC, 31P spectra, etc.). In addition to this critical point, please also address all reviewers' comments point-by-point, including the criticism by reviewer #1 (note that reviewer #3 also provided an attached review file). The quality of the drawings and the NMR peaks assignment in the supporting materials also require extensive improvement before the manuscript can be accepted. The photos of TLCs and chemicals in the flasks should be included in the supporting information rather than in the main manuscript.

Reviewer 1 ·

Basic reporting

The authors previously reported the catching-by-polymerization (CBP) oligodeoxynucleotide (oligo or ODN) purification method, which involves tagging the full-length sequences with a methacrylamide polymerizable tagging phosphoramidite (PTP) in the last cycle of automated oligo synthesis. The full-length sequences are co-polymerized into the gel, allows facile separation from failure sequence and impurity by washing. The full-length products are cleaved from the polymer by acid-treatment. Synthetic procedure of corresponding PTP has been reported previously (Green Chem., 2016, 18, 1125) by column chromatography. In this report, they synthesized several precursors without column chromatography. Synthetic procedures are described.

1) Clear and unambiguous etc.. OK
2) Literature references... OK
3) Professional article structure... very poor. Quality of figures is very low.

4) Self-contained with... ???

Experimental design

I agree that chromatography-free synthesis certainly contributes to lowering the cost. But the synthetic method disclosed here was minor change of the previously reported one.

1) Research question well defined... poor
2) Rigorous inestigation ... poor
3) Methods described... average

Validity of the findings

Since I could not see new findings that were worth reporting, I am rather negative to accept this manuscript.

1) Meaningful replication... poor
2) All underlying... Partly OK
3) Conclusion... ??

Reviewer 2 ·

Basic reporting

The manuscript describes the organic-chemical synthesis of a DNA building block carrying a methylacrylamide group for later polymerization. The latter is used to purify the synthesized DNA; this method, however, is not part of this manuscript. It is just the motivation to publish the synthesis of this special DNA building block. The synthesized DNA building block is novel and should be published. But I have serious doubts about the broad utilization of this method in the future. Some of the synthetic steps are not easy and as long as the new phosphoramidite is not commercially available at low price, nearly nobody will use this method.
For the readers it would be important to compare the scalability and the finally reached purity of the oligonucleotides by HPLC (analytical, semi-preparative and preparative), PAGE and Catching-by-Polymerization.

Experimental design

The experimental design is good and well explained. It fits into the scope of the journal.

Validity of the findings

The synthesis is well described. However, the authors do not provide any proof of purity for the synthesized compounds. They just write that "NMR and MS data are not given here because all the compounds are known. It is crucial here to provide the data for the compound batches described herein, especially with respect to the fact, that the authors had the goal to do the syntheses without flash chromatography and on larger scale. A proof of purity with experimental error must be provided. The compounds might not be new, but the way to prepare it.

Additional comments

Figure 2 does not fit the criteria of a professional figure. The t.l.c images are hard to read and the right figure just shows the foam of a colorless compounds. I think this figure can be removed completely.

Reviewer 3 ·

Basic reporting

Majority of the attributes are covered in the manuscript. Recommended changes are listed below and in the final review.

1. Figure 2 is already in SI, therefore, no need to include in the main text of the manuscript. The observation of clean reaction profile is already summarized along with isolation protocols.

2. The authors should include references to other work which is relevant to the molecules described herein.

Experimental design

The main focus of this work is to offer an efficient and low-cost synthesis protocol for key building blocks previously difficult to make. It would be useful to compare and show how much of improvement is accomplished with new protocols. For example, indicate overall improvement in % yield for the production of compounds 10a-d.

Validity of the findings

The authors make a comment that they have used these amidites to assemble 400 mer long sequence and purified it. It would be useful to add few lines about the application of these amidites and benefit it may offer in harvesting purified oligonucleotides.

Additional comments

Conceptually, this protocol is applicable to purification of RNA sequences too. Given the rising interest in synthesis of siRNA, authors should state if they have used their technique for purification of synthetic RNA or not.

Annotated reviews are not available for download in order to protect the identity of reviewers who chose to remain anonymous.

---

## Round 0.2 · accepted · Accept

Based on your responses and the reviewer's recommendation, your manuscript is ready for publication subjecting to minor modifications that could be done at the proof stage:
- Affiliation: Since all authors share the same affiliation, the affiliation identification number (designated as 1) is not required.
- ref 1: 10.26434/chemrxiv-22024-jcn26439 -> 10.26434/chemrxiv-2024-jcn9d
- ref 26: 10.26434/chemrxiv-22024-zb26437vk -> 10.26434/chemrxiv-2024-zb7vk

Reviewer 3 ·

Basic reporting

The references 1 and 26 appear to be incorrect when searched on the web for the articles. Please correct and offer a link that takes the readers to the original work available on web under ChemRxiv.

Experimental design

The revised protocols are adequate for reproducibility of the experiments and synthesis of products isolated without chromatography.

Validity of the findings

The revised and updated text is clear and sufficient for understanding the scope of this study.

Additional comments

The revised manuscript is now appropriate for publication after references are corrected.